# Measuring activity engagement in old age: An exploratory factor analysis

**Calum Marr** , **Eleftheria Vaportzis**, **Malwina A. Niechcial** , **Michaela Dewar, Alan J. Gow***

Department of Psychology and Centre for Applied Behavioural Sciences, Heriot-Watt University, Edinburgh, United Kingdom

¤ Current address: Division of Psychology, University of Bradford, Bradford, United Kingdom
* a.j.gow@hw.ac.uk

**Data Availability Statement:** The dataset used for the present study (inclusive of the minimal dataset necessary to replicate the findings presented) cannot be deposited in a public data repository for

## Abstract

A growing body of literature suggests that higher engagement in a range of activities can be beneficial for cognitive health in old age. Such studies typically rely on self-report questionnaires to assess level of engagement. These questionnaires are highly heterogeneous across studies, limiting generalisability. In particular, the most appropriate domains of activity engagement remain unclear. The Victoria Longitudinal Study-Activity Lifestyle Questionnaire comprises one of the broadest and most diverse collections of activity items, but different studies report different domain structures. This study aimed to help establish a generalisable domain structure of the Victoria Longitudinal Study-Activity Lifestyle Questionnaire. The questionnaire was adapted for use in a sample of UK-based older adults (336 community-dwelling adults aged 65–92 with no diagnosed cognitive impairment). An exploratory factor analysis was conducted on 29 items. The final model retained 22 of these items in a six-factor structure. Activity domains were: Manual (e.g., household repairs), Intellectual (e.g., attending a public lecture), Games (e.g., card games), Religious (e.g., attending religious services), Exercise (e.g., aerobics) and Social (e.g., going out with friends). Given that beneficial activities have the potential to be adapted into interventions, it is essential that future studies consider the most appropriate measurement of activity engagement across domains. The factor structure reported here offers a parsimonious and potentially useful way for future studies to assess engagement in different kinds of activities.

## Introduction

The potential cognitive benefits of an active lifestyle in old age are well documented. The 'use it or lose it' hypothesis holds that the brain can be 'kept fit' through engagement in stimulating activities [1]. Potentially stimulating activities are often categorised into three broad areas: mentally engaging activities (e.g., reading books or solving crosswords; also commonly referred to as cognitively engaging or intellectually engaging), physically engaging activities (e.g., exercise or sport) and socially engaging activities (e.g., visiting friends or relatives). Engagement in these areas has been linked to more favourable cognitive outcomes [2–4].

legal and ethical reasons. Study participants did not provide written consent for their data to be made publicly available. Furthermore, the dataset contains potentially identifying or sensitive information and making data publicly available could pose privacy concerns. Participants provided written consent for their anonymised data to be used for future ethically approved studies. Any requests for access to any data related the present study should be addressed to Professor Alan J. Gow, a.j.gow@hw.ac.uk, or the Research Engagement Directorate at Heriot-Watt University via purehelp@hw.ac.uk, who oversee open research practices.

**Funding:** This work was supported by Velux Stiftung, Switzerland, https://veluxstiftung.ch, Project No. 1034 to A.J.G. and Project No. 1034a to A.J.G. and M.D. The funders had no role in study design, data collection and analysis, decision to publish, or preparation of the manuscript.

**Competing interests:** The authors have declared that no competing interests exist.

Understanding these associations is an important step in the design and implementation of interventions to promote healthy cognitive ageing. It is therefore important that researchers have conceptually and psychometrically sound measures to assess activity engagement.

Bielak [5, 6] notes that when assessing the association between activity engagement and cognition in old age, a fundamental question remains: what is the best way to measure engagement? Researchers exploring a broad range of possible activities often employ self-report questionnaires. Typically, individuals are presented with a list of activities and asked to indicate how often they normally engage in them. This approach provides a simple, efficient way to assess activity engagement. However, questionnaires vary across studies in several ways, including number of items (i.e., activities) and response options (e.g., yes/no dichotomy vs rating scales).

Another major source of variation across studies is the number of activity domains assessed. While some studies derive a single measure of overall activity engagement, e.g., [7], others group items to provide scores assessing engagement in specific 'types', or domains, of activity. For example, some studies classify activities into the three conventional domains of mental, physical and social [8]. While useful for discursive purposes, this approach can be restrictive in practice, not allowing for the multi-faceted nature of some activities. Other studies include more nuanced activity domains, such as political (e.g., taking part in a demonstration) [9] or artistic (e.g., painting) [10].

There is no current consensus on the most appropriate set of activity domains. Adams et al. [11] found that, across 42 studies, the number of activity domains measured ranged from one to thirteen. Domains may be specified a priori, e.g. [8], or based on post-hoc data reduction techniques such as factor analysis, e.g., [12]. Predefined activity domains, while often based on prior research, can be idiosyncratic. For example, Wang et al.'s measure of mental engagement includes activities such as sewing or attending an opera [8], while Verghese et al.'s includes activities such as writing and group discussions [13]. Data reduction techniques such as factor analysis can provide a more empirical approach to measuring specific activity domains, although associations among items may be specific to the analytic sample; validation of a factor structure across samples is necessary for generalisability.

The diversity of methods in the literature can be illustrated by a specific example: the Victoria Longitudinal Study-Activity Lifestyle Questionnaire (VLS-ALQ). First described in Hultsch et al. [14], the full questionnaire asks individuals to rate their level of engagement in 70 activities. Based on a content analysis of these items, the authors proposed six activity domains (Physical, Social, Self-Maintenance, Passive Information Processing, Integrative Information Processing and Novel Information Processing). However, other studies utilising this questionnaire have used different domains, with the rationale for the categorisation of items often unreported. For example, Hultsch et al. [15] included a domain measuring engagement in hobbies and home maintenance activities, while Small et al. [16] instead classified items into the three broad domains of mental, physical and social activities. Jopp et al. [17] modified the questionnaire by removing items related to self-maintenance and adding new items related to physical and social activities. The authors validated an eleven-factor structure, categorising items into the domains of Physical, Crafts, Games, Watching TV, Social-Private, Social-Public, Religious, Technology Use, Developmental, Experiential and Travel.

The large pool of items within the VLS-ALQ make it a good candidate for investigating activity engagement. However, variation in the factor structure adopted across studies highlights the central issue of making generalisations; it is difficult to synthesise results when studies using the same instrument adopt differing approaches. This is magnified across the literature when the fuller diversity of measures is considered. The present study examined the factor structure of an adapted version of the VLS-ALQ with a UK-based sample of older adults.

By exploring the congruence of a factor structure derived from a geographically distinct sample, this study was intended to contribute to a consensus on the underlying dimensionality not only of the VLS-ALQ, but also of activity engagement more broadly.

## Materials and methods

### Participants

Participants were recruited as part of an activity-based intervention study. Eligible participants had to be aged 65 or over and fluent in English. Exclusion criteria were a) diagnosis of a neuro-degenerative condition or brain tumour; b) stroke; c) head injury; d) significant manual impairment; e) prior engagement in all the activities offered as part of the intervention study; f) living outside the study recruitment area and being unable to travel to the area regularly. Participants were recruited via volunteer databases, flyers, local newspaper adverts, contact with local community groups and social media.

A total of 416 individuals were screened for eligibility. Eighty did not meet the criteria and were excluded or withdrew from the study after screening. The remaining 336 volunteers were enrolled on the study and comprise the analytic sample (102 men, 234 women; for sample characteristics see Table 1). All participants lived in Scotland, predominantly around Edinburgh and the Lothians. Three hundred and thirty participants (98.2%) identified as White (British/Irish/European/Other White Background). All data reported here were collected as part of participants' baseline assessments, prior to intervention group allocation (assessments were conducted between June 2017 and December 2019). All participants provided written informed consent at the start of their assessment. The study received ethical approval from the Heriot-Watt University School of Social Science Ethics Committee (Ref: 2017–453) and the NHS South East Scotland Research Ethics Committee 02 (REC Ref: 17/SS/0153; SSA Ref: 17/SS/0157).

### Measures and procedure

Participants completed an adapted version of the 57-item VLS-ALQ validated by Jopp et al. [17]. Each item comprises a statement (e.g., "I read newspapers") with nine response options to indicate current frequency of engagement in the activity. Response options are coded as follows: 0 = Never, 1 = Less than once a year, 2 = About once a year, 3 = Two or three times a year, 4 = About once a month, 5 = Two or three times a month, 6 = About once a week, 7 = Two or three times a week, 8 = Daily. The present study adapted this version in two ways. First, the wording of some of the items was amended from North American English to UK English. Second, three items from the original 70-item questionnaire [14] were re-included

**Table 1. Sample characteristics.**

| Variable | Mean | SD | Min | Max | Skew | Kurtosis |
|---|---|---|---|---|---|---|
| Age | 71.4 | 5.4 | 65 | 92 | 1.00 | 0.67 |
| Years of Education | 15.9 | 3.5 | 9 | 26 | 0.03 | -0.51 |
| Deprivation | 16.2 | 4.6 | 2 | 20 | -1.23 | 0.53 |
| Self-Rated Health[a] | 3.7 | 0.9 | 1 | 5 | -0.47 | -0.13 |
| MMSE Total Score | 28.9 | 1.2 | 23 | 30 | -1.62 | 3.55 |

*Note.* Deprivation = Scottish Index of Multiple Deprivation (vigintile ranking [i.e. 1–20] based on postcode—lower values indicate higher deprivation). MMSE = Mini-Mental State Examination; scores range from 0 to 30—lower values indicate cognitive problems.

[a] 1 = Poor, 2 = Fair, 3 = Good, 4 = Very Good, 5 = Excellent.

(listen to radio, attend concert/play, attend sports events), along with two new items covering activities not included in the original VLS-ALQ (go to galleries/museums, go to pubs/social clubs), leading to a total of 62 items.

The short version of the International Physical Activity Questionnaire (IPAQ) [18] was used to assess participants' current level of physical activity. The questionnaire asks individuals to record the amount of time they spent engaging in vigorous activity, moderate activity and walking respectively over the preceding week. This information was used to calculate a measure of total metabolic equivalent of task (MET) minutes-per-week (i.e., overall energy expenditure; test-retest reliability = .76) [18].

Years of education was recorded as the total number of years in primary, secondary and further/higher education. The Scottish Index of Multiple Deprivation 2016 (SIMD16) [19] was used as a proxy measure of socioeconomic status. Geographical areas are ranked according to overall level of deprivation using a combination of multiple indicators (e.g., income, education, housing) and rankings are split into vigintiles (1 = most deprived to 20 = least deprived). Vigintile rankings for each participant were obtained based on their postcodes.

As an indicator of perceived health-status, participants were asked to rate their general health on a five-point scale (1 = Poor, 2 = Fair, 3 = Good, 4 = Very Good, 5 = Excellent). The Mini-Mental State Examination (MMSE) [20] was used as an indicator of global cognitive status. The MMSE is a standardised instrument that is widely used to detect possible cognitive impairment in both clinical and research settings. The test takes around five minutes to administer, with questions covering orientation (time and place), memory (registration and recall of three words), concentration (spelling 'world' backwards), language (naming objects, sentence repetition and reading comprehension) and praxis (ideational [folding a piece of paper in half], copying/drawing and spontaneous writing). Scores range from 0–30; test-retest reliability estimates typically range from .80 -.95 [21]. MMSE scores $\leq$ 23 are often taken to indicate potential cognitive impairment. One participant scored below this cut-off. However, MMSE was not used as an exclusion criterion in the present study in order to avoid excluding individuals with low education (MMSE score has been shown to be sensitive to level of education) [22].

Assessments were conducted in person by a member of the research team and normally lasted for 3 to 3.5 hours. Participants provided demographic information and completed a battery of cognitive and physical tests, including the MMSE, over the course of their assessment. All members of the research team were trained in the administration of these standardised tests prior to the commencement of data collection. At the end of the assessment, participants were given a booklet of questionnaires, including the VLS-ALQ and the IPAQ, to complete at home and return to the research team by post. In the case that a questionnaire was returned with missing responses, attempts were made to contact the participant via phone or email to request their answers.

## Statistical analysis

Where missing data remained, this was dealt with via multiple imputation (MI). The MI process involves the replacement of missing values with a set of $m > 1$ plausible values, resulting in $m$ complete sets of data containing varying estimates of the missing values. Standard statistical methods are then applied to each of these datasets, and parameter estimates and standard errors are pooled to form a single set of results according to formulae suggested by Rubin [23]. In comparison to more conventional approaches to missing data (e.g., listwise deletion), multiple imputation yields unbiased parameter estimates when data are missing at random (i.e., missingness in a given variable does not depend on the variable itself, but may depend on

other variables) [24]. For a full summary of the missing data and multiple imputation process, see S1 Appendix.

All analyses were conducted using R (version 4.0.1) [25]. Exploratory factor analysis (EFA) was used to examine the underlying factor structure of the adapted version of the VLS-ALQ. EFA was used as the aim of the analysis was to identify true latent domains that explained common variance among the activity items rather than simply summarise the maximum amount of observed variance, as is done in principal component analysis [26]. As suggested by Nassiri et al. [27], the EFA was conducted on the pooled Pearson correlation matrix of all imputed datasets (Pearson correlations have been recommended as appropriate for use when ordinal variables have five or more response categories and are approximately normally distributed) [26]. A combination of parallel analysis [28] and Velicer's Minimum Average Partial (MAP) test [29] was used to establish the possible number of factors. Parallel analysis on the pooled correlation matrix has been shown to perform well in determining the correct number of factors [30]. The number of factors suggested by parallel analysis was used as a maximum estimate, while MAP was used to indicate a minimum; all possibilities in between were investigated.

The fa() function in the psych package (version 1.9.12) [31] was used to estimate models. Models were estimated using maximum likelihood with oblique (oblimin) rotation. Given the distributional assumptions of maximum likelihood estimation, items that were substantially non-normal (absolute univariate skew > 2 and/or absolute univariate kurtosis > 7) [32] across all imputations were excluded prior to conducting the analysis. Items that did not correlate at a magnitude of at least .3 with one or more other items were also excluded [33]. Finally, items with a Kaiser-Meyer-Olkin (KMO) measure of sampling adequacy value of less than 0.5 (the bare minimum proposed by Kaiser) [34] were excluded. Models were then estimated on the remaining pool of items. Bartlett's test was conducted to ensure that correlations between these items were of sufficient size for EFA. The determinant of the item correlation matrix was calculated to assess multicollinearity among the items (a value > .00001 can be taken as indicating an absence of multicollinearity) [35].

For each possible number of factors, models were estimated then moderated in an iterative fashion by excluding items with no pattern coefficients (i.e., loadings) of at least .3 on any factor, and items loading on multiple factors (i.e., items with Hofmann's index of complexity > 2) [36]. Models were deemed viable once all items primarily loaded on one factor and each factor had at least three items loading on it (a minimum that has been suggested as more items per factor improves factor stability) [26, 37]. The most appropriate number of factors was decided by examining and comparing each model's root mean square error of approximation (RMSEA; values < .05 indicating good fit, values .05 - .08 indicating adequate fit) [38], Tucker-Lewis Index (TLI; values ≥ .95 indicating good fit) [39] and conceptual interpretability.

Once the appropriate factor model was selected, activity domain scores were calculated by summing responses to the relevant items for each factor. To explore the validity of these domains, Pearson correlations between domain scores, demographic variables (age, education, deprivation, self-rated health), MMSE scores and total MET-minutes per week were calculated. As there may be gender differences in activity participation [5], Welch's t-tests were also used to compare mean levels of engagement in each domain between men and women.

## Results

Eight items were initially excluded due to highly non-normal distributions (i.e., absolute univariate skew > 2 and/or absolute univariate kurtosis > 7, detailed above). A further 24 items

were excluded due to consistently low correlations with other variables. One item was also excluded due to a low KMO value. Specific reasons for exclusions are listed in S2 Appendix. EFA was conducted on the remaining 29 items.

The overall KMO indicated that the suitability of the data for factor analysis was adequate (KMO = .69). Item-level KMO values ranged from .55 to .79 (see S3 Appendix). Bartlett's test of sphericity, $\chi^2$ (406) = 2020.58, $p < .001$, indicated that item correlations were of an acceptable magnitude for EFA. The determinant of the correlation matrix did not indicate any problem with multicollinearity among the items (determinant = .002).

The MAP test indicated that extracting three factors achieved a minimum average squared partial correlation of 0.01. Parallel analysis suggested that eight factors had eigenvalues greater than would be expected by chance (see S4 Appendix). Thus, all possibilities within the range of three to eight factors were examined.

The eight and seven-factor models failed to meet the outlined criteria for viability (exclusion of items for the reasons outlined above led to some factors being defined by only one or two primary loadings). The six-factor model met the specified criteria after seven items were excluded, the five-factor model did so after ten items were excluded, the four-factor model did so after nine items were excluded and the three-factor model did so after twelve items were excluded.

The six-factor model was found to have the lowest RMSEA value (0.036), indicating good fit (see Table 2). The five and six factor models had the joint highest TLI value (.918; see Table 2), although both were below the cut-off threshold for good model fit of .95 recommended by Hu et al. [39]. Therefore, given that the six-factor model had the lowest RMSEA value and the factors identified in the six-factor model were conceptually sound, this model was selected as the optimal factor structure for the activity items. The pattern matrix for the six-factor model is presented in Table 3. The structure matrix for the six-factor model is presented in S5 Appendix. Pattern/structure matrices for the three, four, five, seven and eight factor solutions are presented in S6 Appendix.

Bolded pattern coefficients presented in Table 3 indicate primary loadings. The items that loaded primarily on the first factor were all related to manual, practical skills (e.g., 'Do household repairs [for example, painting or leaky faucets]', 'Repair a mechanical device [for example, a car or lawn mower]'); this factor was named Manual. The items that loaded primarily on the second factor indicated an interest in learning and culture (e.g., 'Engage in creative writing, writing poems, writing newspaper articles, etc.', 'Read books or magazines as part of my job, career, or formal education'); this factor was named Intellectual. Items such as 'Play card games (for example, Bridge)' and 'Play word games (for example, Scrabble)' loaded primarily on the third factor; this factor was named Games.

The fourth factor was primarily explained by loadings from items associated with religion (e.g., 'Attend church or other religious services', 'Attend organised social events [for example, activities at the community centre or church social groups]'); this factor was named Religious.

**Table 2. Model comparisons.**

| Number of factors | Proportion of variance explained (%) | RMSEA (90% CI) | TLI |
|---|---|---|---|
| 3 | 31.0 | 0.082 (0.072, 0.093) | .700 |
| 4 | 33.1 | 0.070 (0.061, 0.080) | .742 |
| 5 | 39.3 | 0.040 (0.026, 0.053) | .918 |
| 6 | 40.0 | 0.036 (0.023, 0.048) | .918 |

*Note*. RMSEA = root mean square error of approximation. TLI = Tucker-Lewis Index.

**Table 3. Six-factor model pattern matrix and factor correlations.**

| Item | Factor | | | | | | $h^2$ |
| --- | --- | --- | --- | --- | --- | --- | --- |
| | 1. | 2. | 3. | 4. | 5. | 6. | |
| Do household repairs (for example, painting or leaky faucets) | **0.86** | -0.02 | 0.00 | 0.01 | 0.02 | 0.02 | 0.741 |
| Repair a mechanical device (for example, a car or lawn mower) | **0.71** | 0.08 | 0.00 | 0.00 | 0.00 | 0.02 | 0.537 |
| Purchase a new item requiring some set-up or assembly | **0.61** | -0.01 | 0.07 | 0.00 | 0.00 | -0.06 | 0.388 |
| Engage in creative writing, writing poems, writing newspaper articles, etc. | 0.13 | **0.57** | -0.06 | 0.04 | -0.08 | 0.02 | 0.354 |
| Read books or magazines as part of my job, career, or formal education | 0.02 | **0.55** | 0.03 | -0.11 | -0.01 | 0.00 | 0.297 |
| Go to galleries or museums | -0.12 | **0.43** | 0.11 | -0.04 | 0.10 | 0.10 | 0.258 |
| Attend a public lecture or talk | -0.11 | **0.54** | 0.09 | 0.09 | 0.07 | -0.01 | 0.357 |
| Engage in political activities (for example, neighbourhood organisation) | 0.08 | **0.46** | 0.00 | -0.02 | 0.03 | -0.01 | 0.233 |
| Give a public talk or lecture (for example, to a club, service organisation, etc.) | 0.05 | **0.50** | -0.09 | 0.05 | 0.01 | -0.08 | 0.269 |
| Do aerobics (for example, cardiovascular, fitness training, or workout) | 0.04 | -0.01 | 0.03 | 0.06 | **0.65** | -0.02 | 0.429 |
| Do flexibility training (for example, stretching, yoga, or tai chi) | -0.09 | 0.07 | 0.11 | 0.05 | **0.49** | 0.02 | 0.289 |
| Do weight lifting, strength training, or calisthenics | 0.03 | -0.01 | -0.06 | -0.05 | **0.77** | 0.00 | 0.592 |
| Play card games (for example, Bridge) | -0.02 | -0.05 | **0.46** | 0.05 | 0.06 | 0.01 | 0.221 |
| Play board games (for example, chess or checkers) | 0.08 | 0.03 | **0.63** | -0.01 | 0.01 | -0.05 | 0.412 |
| Play knowledge games (for example, Trivial Pursuit) | 0.03 | 0.03 | **0.65** | 0.00 | -0.03 | 0.00 | 0.426 |
| Play word games (for example, Scrabble) | -0.03 | -0.04 | **0.64** | -0.01 | -0.03 | 0.04 | 0.403 |
| Talk on the phone to friends, or relatives | -0.10 | -0.03 | 0.08 | 0.13 | 0.04 | **0.35** | 0.177 |
| Visit relatives, friends, or neighbours | 0.02 | -0.01 | 0.00 | -0.01 | 0.00 | **1.00** | 0.995 |
| Go out with friends | -0.15 | 0.09 | 0.01 | -0.01 | -0.05 | **0.34** | 0.160 |
| Attend church or other religious services | 0.01 | -0.03 | -0.02 | **0.91** | -0.04 | -0.02 | 0.810 |
| Engage in prayer, meditation, or philosophical contemplation | -0.01 | 0.08 | 0.06 | **0.54** | 0.11 | 0.00 | 0.343 |
| Attend organised social events (for example, activities at the community centre or church social groups) | -0.05 | 0.13 | 0.02 | **0.47** | 0.05 | 0.13 | 0.297 |
| Sum of Squared Loadings | 1.73 | 1.61 | 1.50 | 1.39 | 1.30 | 1.28 | |
| Variance Explained (%) | 7.9 | 7.3 | 6.8 | 6.3 | 5.9 | 5.8 | |
| Factor Correlations | | | | | | | |
| 1. Manual | - | - | - | - | - | - | |
| 2. Intellectual | .19 | - | - | - | - | - | |
| 3. Games | .09 | .20 | - | - | - | - | |
| 4. Religious | -.03 | .19 | .09 | - | - | - | |
| 5. Exercise | .15 | .26 | .11 | .00 | - | - | |
| 6. Social | -.10 | .09 | .14 | .05 | .01 | - | |

*Note*. **Bold** = primary loadings (pattern coefficient > .3); $h^2$ = communality. VLS-ALQ items are included with permission to support the analyses; permission to use the VLS-ALQ in full or in part must be obtained from Professor Roger Dixon (rdixon@ualberta.ca).

Items related to physical activity loaded primarily on the fifth factor (e.g., 'Do aerobics [for example, cardiovascular, fitness training, or workout]', 'Do weight lifting, strength training or calisthenics'); this factor was named Exercise. The sixth factor was primarily explained by a very large loading of 'Visit relatives, friends or neighbours' and two less substantial loadings of 'Talk on the phone to friends or relatives' and 'Go out with friends'; this factor was named Social. Correlations between the factors were generally small, ranging in magnitude from -.10 to .26 (see Table 3).

For comparison, a complete-case analysis was also conducted including only those participants who had complete data for the 22 items in the six-factor solution (N = 311). The six factors extracted were conceptually identical to those reported above, and the pattern of loadings was the same (see S7 Appendix). The model explained 43.7% of the variance and fit indices

**Table 4. Activity domain scores: Gender comparisons and correlations with demographic variables, MMSE scores and total MET-minutes per week.**

| Activity Domain | Mean (SD) Men | Mean (SD) Women | t | df | p | Correlations | | | | | |
|---|---|---|---|---|---|---|---|---|---|---|---|
| | | | | | | Age | Years of Education | Deprivation | Self-Rated Health[a] | MMSE | Total MET-minutes per week |
| Manual | **8.49 (4.97)** | **3.89 (3.83)** | **8.94** | **107.09** | **<0.001** | -.08 | .14* | -.10 | -.01 | -.11 | .12 |
| Intellectual | 11.81 (8.25) | 11.11 (7.91) | 0.73 | 125.20 | 0.469 | -.08 | .30*** | .00 | .05 | .00 | .14* |
| Games | 5.96 (6.90) | 7.23 (6.73) | -1.72 | 95.61 | 0.089 | -.06 | .02 | -.05 | -.01 | -.05 | .12 |
| Religious | **5.78 (6.06)** | **7.86 (6.28)** | **-2.95** | **175.60** | **0.004** | .10 | -.04 | .03 | -.02 | -.05 | .02 |
| Exercise | 6.69 (7.72) | 6.93 (6.72) | -0.28 | 129.84 | 0.780 | -.01 | .12* | .01 | .04 | -.13 | .26*** |
| Social | **15.24 (4.12)** | **17.33 (3.82)** | **-4.49** | **71.61** | **<0.001** | .06 | -.01 | .10 | .11 | .08 | .05 |

*Note*. Deprivation = Scottish Index of Multiple Deprivation (vigintile ranking [i.e. 1–20] based on postcode—lower values indicate higher deprivation). MMSE = Mini-Mental State Examination (scores range from 0 to 30—lower values indicate cognitive problems). MET = metabolic equivalent of task; total MET-minutes per week measured using the International Physical Activity Questionnaire (higher scores indicate higher levels of overall physical activity). Significant gender differences (indicated by boldface) remained so after Bonferroni correction.

[a]1 = Poor, 2 = Fair, 3 = Good, 4 = Very Good, 5 = Excellent.

*p < .05

**p < .01

***p < .001.

were largely comparable (RMSEA = 0.048 [90% CI = 0.037, 0.060], TLI = .880). For descriptive purposes, omega reliability coefficients were calculated for each factor using the complete cases. Reliability coefficients were as follows: Manual ($\omega$ = 0.85), Intellectual ($\omega$ = 0.69), Games ($\omega$ = 0.69), Religious ($\omega$ = 0.72), Exercise ($\omega$ = 0.66), Social ($\omega$ = 0.65).

Mean comparisons between men and women on activity domain scores are presented in Table 4. On average, men scored significantly higher than women in the Manual domain, while women scored significantly higher in the Religious and Social domains. There were no significant correlations between any of the activity domain scores and either age, deprivation, self-rated health or MMSE score (see Table 4). Years of education demonstrated small but significant positive correlations with scores in the Manual ($r$ = .14, $p < .05$), Intellectual ($r$ = .30, $p < .001$) and Exercise ($r$ = .12, $p < .05$) domains. Scores in the Exercise domain were significantly positively correlated with total MET-minutes per week from the IPAQ ($r$ = .26, $p < .001$), as were scores in the Intellectual domain ($r$ = 0.14, $p < .05$).

## Discussion

The VLS-ALQ has been utilised by several well-cited studies investigating the association between activity engagement and cognition in older age [14–16]. However, the domains used to summarise activity engagement have varied across these studies. The present study adapted a modified version of the VLS-ALQ [17] for use in a UK-based sample of older adults. An exploratory factor analysis identified six activity domains from 22 items: Manual, Intellectual, Games, Religious, Exercise and Social.

Jopp et al. [17] validated an eleven-factor structure across two large samples using their 57-item version of the questionnaire. These samples included adults of all ages, unlike the older sample utilised in the present study. However, there are some notable consistencies across both studies. The items that load on the Manual, Games, Religious, Exercise and Social factors in the present study also loaded on conceptually similar factors in Jopp et al.'s study (termed Crafts, Games, Religious, Physical and Social-Private respectively) [17]. Only the Intellectual factor identified here is theoretically distinct. The items comprising this factor

were instead categorised as either Developmental ('Engage in creative writing', 'Read books or magazines as part of my job', 'Attend a public lecture or talk') or Social-Public ('Engage in political activities', 'Give a public talk or lecture') by Jopp et al. 'Go to galleries or museums', which also loaded on the Intellectual factor, was a new item added in the present study. Collectively, these items indicate an interest in learning and culture, something that has not been fully captured in previous uses of the VLS-ALQ.

The present results reinforce the importance of physical and social activity as theoretically meaningful domains of activity engagement. It should be noted, however, that the physical and social domains reported here have more conceptual specificity than those of other questionnaires. The physical activity domain in this study was named was Exercise as the three relevant items characterise the type of individual physical activity that one might engage in for the specific purpose of maintaining fitness (aerobics, flexibility training and weightlifting). This is distinct from other questionnaires that also include items measuring engagement in competitive sports, outdoor activities or gardening [8, 9, 13]. Further research will be necessary to explore whether this qualitative distinction affects any association between physical activity and cognitive outcomes. The Social domain in the present study is similarly distinct, focusing more on socialising with a close group of friends and relatives. This is similar to the measure of 'one-on-one' social activity in Bielak et al. [40], which, unlike group social activity (i.e., clubs), was associated with changes to cognitive performance among older adults.

There is also evidence of the Manual, Games and Religious domains in other activity questionnaires. The Manual domain reported here could be considered analogous to the Realistic domain reported in Parslow et al. [10], which was defined by items such as 'Fixed mechanical things' and 'Used metalwork or machine tools'. Ghisletta et al. [12] identified an activity factor similar to Games that they termed Leisure (defined by two items: 'play games' and 'do crossword puzzles'). The same study also identified a similar Religious factor (items: 'pray' and 'attend religious services').

Some evidence of construct validity was found. Of particular note was the significant positive correlation ($r = .26$, $p < .001$) between scores in the Exercise domain and total MET-minutes per week from the IPAQ (a broader measure of overall physical activity). As noted above, the Exercise domain of the VLS-ALQ measures frequency of engagement in specific fitness-related activities, while the measure of MET-minutes per week from the IPAQ considers the *intensity* of physical exertion across a wider range of activities. This may explain that while the observed correlation was in the expected direction, it was small in magnitude. The observed tendencies for men to engage in more manual activities and for women to engage in more social activities were also reported by Parslow et al. [10]. There was no evidence of any relationship between activity domain scores and a measure of cognitive function (MMSE score). This may be due to a ceiling effect–as would be expected in a sample of adults without diagnosed cognitive impairments, most participants scored towards the top end of the scale.

The activity domains reported here do not necessarily represent the definitive structure of the VLS-ALQ, or of activity engagement more broadly. The use of a small convenience sample is a limitation of the present study and it is important to acknowledge that results may have been influenced by the self-selecting nature of this sample. Participants were almost all White, generally healthy, highly educated and lived in less deprived areas–such individuals are likely to behave in certain, relatively consistent ways and as such variability within the sample may have been limited. This issue is highlighted by the fact that several items were excluded from the analysis due to highly skewed distributions.

Many of the items were also excluded due to low correlations with other items. This was done in order to keep only those items that shared a sizable amount of common variance, which is an important condition for factor analysis [33]. However, excluding these items may

explain why the present analysis did not identify four of the domains reported by Jopp et al. [17]. Items that loaded on the factors termed Watching TV, Travel, Experiential and Technology Use in Jopp et al.'s study were excluded in the present study due to non-normal distributions or low correlations with other items. Interestingly, the authors note that these four factors exhibited lower reliability estimates relative to others, suggesting that they may have poor generalisability. In other words, while the approach adopted here led to the initial exclusion of a large proportion of items, this may have only resulted in the loss of factors with poor reliability, while the most reliable factors identified by prior research were still largely replicated.

The exploratory nature of this analysis is also a limitation. Ideally, the factor structure here would be replicated in an independent sample using confirmatory factor analysis. The small sample size in the present study precluded splitting the sample and performing both exploratory and confirmatory analyses. Attempts to replicate these findings in future studies using the VLS-ALQ or another questionnaire are therefore strongly encouraged. Further attempts to identify other potentially important domains that were missed here due to sample limitations are also encouraged. Such studies could use larger, more diverse samples; as Bielak [5] notes, cultural background, socioeconomic status and geographic location are all factors that can influence the kinds of activity one engages in. Achieving a generalisable model of the dimensionality of activity engagement will require careful consideration of these factors.

Unlike other questionnaires, e.g., [6], the VLS-ALQ only measures frequency of engagement and does not consider the relative level of mental, physical or social demand of each activity. However, it is reasonable to assume that the Exercise and Social domains are predominantly physically and socially demanding respectively. The Religious domain appears to be mainly socially demanding, although it also includes more solitary spiritual activity like prayer or mediation. Items that loaded on the Intellectual domain (e.g., writing, reading) and the Games domain (e.g., word games, card games) are commonly used to measure mental engagement [13, 41, 42]. Another measure of mental of mental engagement also includes items similar to those that load on the Manual domain (e.g., household repairs) [43]. Therefore, the Intellectual, Games and Manual domains could each be viewed as particularly mentally demanding. More research is necessary to establish this empirically.

The purpose of this study was to explore what the most appropriate domains of activity engagement were in an existing questionnaire. In doing so, this study aimed to highlight the growing need for consensus in how activity engagement is measured, particularly in the field of cognitive ageing. The domain structure reported here could provide a foundation for a more consistent approach across future studies investigating the cognitive benefits of activity engagement. As highlighted above, the domains identified in this study offer more conceptual specificity than the broader domains of mental, physical or social activity. Future studies could explore whether these new factors are useful in predicting cognitive ability or cognitive change in older adults. For example, studies could examine whether engagement in one or more of the domains identified here is associated with performance in specific cognitive domains.

In order to make generalisations about the effects of certain types of activity on cognition in older age, greater consideration must be given to the assessment of activity engagement itself. Despite the limitations outlined above, the present findings support the usefulness of the VLS-ALQ as a measure of meaningful areas of engagement. Five of the factors identified were conceptually similar to those reported in other studies, suggesting that manual/craft activities, games, religious activities, exercising and socialising represent consistent underlying patterns of activity engagement in older adults. An additional, novel factor (Intellectual) was also identified, suggesting certain older adults may have a specific propensity for activities involving learning and culture. While further validation is imperative, the activity factors described here could offer a useful framework for measurement of activity engagement in older age.

## Supporting information

**S1 Appendix. Summary of multiple imputation process.**
(DOCX)

**S2 Appendix. Items excluded prior to exploratory factor analysis.**
(DOCX)

**S3 Appendix. Summary statistics for 29 items included in exploratory factor analysis.**
(DOCX)

**S4 Appendix. Parallel analysis scree plot.**
(DOCX)

**S5 Appendix. Six-factor model structure matrix.**
(DOCX)

**S6 Appendix. Pattern/structure matrices for alternative factor solutions.**
(DOCX)

**S7 Appendix. Complete case analysis: Six-factor model pattern/structure matrices.**
(DOCX)

## Acknowledgments

The authors would like to thank Christopher Hertzog and Daniela Jopp for sharing their adapted version of the Victoria Longitudinal Study-Activity Lifestyle Questionnaire (VLS-ALQ), and Roger Dixon for approval to include the items in the current manuscript (please note, the items are included here to support the analyses; permission to use the VLS-ALQ in full or in part must be obtained from rdixon@ualberta.ca). We would like to thank Andrew Pearce for assisting with data checking. We would also like to thank our volunteer study participants, without whom this research would not have been possible.

## Author Contributions

**Conceptualization:** Alan J. Gow.

**Data curation:** Calum Marr, Eleftheria Vaportzis, Malwina A. Niechcial.

**Formal analysis:** Calum Marr.

**Funding acquisition:** Michaela Dewar, Alan J. Gow.

**Investigation:** Calum Marr, Eleftheria Vaportzis, Malwina A. Niechcial.

**Methodology:** Eleftheria Vaportzis, Malwina A. Niechcial, Alan J. Gow.

**Project administration:** Calum Marr, Eleftheria Vaportzis, Malwina A. Niechcial, Alan J. Gow.

**Supervision:** Michaela Dewar, Alan J. Gow.

**Writing – original draft:** Calum Marr.

**Writing – review & editing:** Calum Marr, Eleftheria Vaportzis, Malwina A. Niechcial, Michaela Dewar, Alan J. Gow.

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
