## [Decision Letter · Decision Letter 0]

21 Sep 2021

PONE-D-21-25224Measuring activity engagement in old age: an exploratory factor analysisPLOS ONE

Dear Dr. Gow,

Thank you for submitting your manuscript to PLOS ONE. After careful consideration, we feel that it has merit but does not fully meet PLOS ONE’s publication criteria as it currently stands. Therefore, we invite you to submit a revised version of the manuscript that addresses the points raised during the review process.

ACADEMIC EDITOR: Please insert comments here and delete this placeholder text when finished. Be sure to:Indicate which changes you require for acceptance versus which changes you recommendAddress any conflicts between the reviews so that it's clear which advice the authors should followProvide specific feedback from your evaluation of the manuscriptPlease ensure that your decision is justified on PLOS ONE’s publication criteria and not, for example, on novelty or perceived impact.

We look forward to receiving your revised manuscript.

Kind regards,

Mohammad Asghari Jafarabadi

Academic Editor

PLOS ONE

Journal Requirements:

Additional Editor Comments (if provided):

Reviewers' comments:

Reviewer's Responses to Questions

**Comments to the Author**

1. Is the manuscript technically sound, and do the data support the conclusions?

Reviewer #1: Yes

Reviewer #2: Yes

2. Has the statistical analysis been performed appropriately and rigorously? 

Reviewer #1: Yes

Reviewer #2: Yes

3. Have the authors made all data underlying the findings in their manuscript fully available?

Reviewer #1: No

Reviewer #2: No

4. Is the manuscript presented in an intelligible fashion and written in standard English?

Reviewer #1: Yes

Reviewer #2: Yes

5. Review Comments to the Author

Reviewer #1: The reviewed paper discusses inconsistency in the definition of activity engagement domains in studies of cognitive health in advanced age. To this end, the authors performed an exploratory factor analysis on data collected from adults age 65+ in the UK using an adapted version of the VLS-ALQ.

I commend the authors for their effective communication of the rationale, methods, and results of their work. The methods were particularly well-described and justified, which is important given the authors encourage testing these study findings in different populations. While this research alone cannot create census on what domains are most appropriate (which the authors themselves point out), it does lay a good methodological foundation for how those domains should be defined. This foundation could, in turn, lead to greater consistency when classifying domains for studies on cognitive health in advanced age.

There are several strong limitations of this study, but I believe they are adequately covered by the authors in the discussion section.

At line [174] I would like to know if the research team members received standardized training before administering the assessments.

Within the limits of my expertise, I think this is solid work and should be accepted for publication.

Reviewer #2: Review of the manuscript titled: Measuring activity engagement in old age: an exploratory factor analysis

Dear Editor

Thank you for allowing me to review the manuscript for this Research Article

1. The language, flow and the content of this paper are well written.

2. Materials and Methods line 117 to 119 not captured in Table 1. I recommend this information to be added in the same table.

3. In statistical analysis part, I recommend to show how you addressed the multicollinearity of different domains/items.

4. In the result section: Line 233- As an exploration factor analysis it is important to show the distribution figure (to represent different pattern; normal and non-normal, skewness, kurtosis)

5. Discussion: Line 371-372 include this (r = .26, p < .001) to support your discussion that there was a significant positive correlation between scores in the Exercise domain and total MET-minutes per week from the IPAQ.

6. Finally, I’m pretty convinced to include as an appendix or a link for the Victoria Longitudinal Study-Activity Lifestyle Questionnaire (VLS-ALQ) in this manuscript.

6. PLOS authors have the option to publish the peer review history of their article (what does this mean?). If published, this will include your full peer review and any attached files.

Reviewer #1: No

Reviewer #2: **Yes: **Dr. Dennis M. Lyakurwa

---

## [Decision Letter · Decision Letter 1]

22 Nov 2021

Measuring activity engagement in old age: an exploratory factor analysis

PONE-D-21-25224R1

Dear Dr. Gow,

We’re pleased to inform you that your manuscript has been judged scientifically suitable for publication and will be formally accepted for publication once it meets all outstanding technical requirements.

Kind regards,

Mohammad Asghari Jafarabadi

Academic Editor

PLOS ONE

Reviewers' comments:

Reviewer's Responses to Questions

**Comments to the Author**

1. If the authors have adequately addressed your comments raised in a previous round of review and you feel that this manuscript is now acceptable for publication, you may indicate that here to bypass the “Comments to the Author” section, enter your conflict of interest statement in the “Confidential to Editor” section, and submit your "Accept" recommendation.

Reviewer #1: All comments have been addressed

Reviewer #2: All comments have been addressed

2. Is the manuscript technically sound, and do the data support the conclusions?

Reviewer #1: Yes

Reviewer #2: Yes

3. Has the statistical analysis been performed appropriately and rigorously? 

Reviewer #1: Yes

Reviewer #2: Yes

4. Have the authors made all data underlying the findings in their manuscript fully available?

Reviewer #1: No

Reviewer #2: Yes

5. Is the manuscript presented in an intelligible fashion and written in standard English?

Reviewer #1: Yes

Reviewer #2: Yes

6. Review Comments to the Author

Reviewer #1: I appreciate the care that the authors put into responding to and addressing reviewer comments. I have no additional concerns.

Reviewer #2: Dear Authors

I acknowledge to receive your responses.

Your responses were clear and relevant to address my questions

I wish you a successful publication

See you in next work

7. PLOS authors have the option to publish the peer review history of their article (what does this mean?). If published, this will include your full peer review and any attached files.

Reviewer #1: No

Reviewer #2: **Yes: **Dennis Modestus Lyakurwa

---

## [Editor Report · Acceptance letter]

26 Nov 2021

PONE-D-21-25224R1 

Measuring activity engagement in old age: an exploratory factor analysis 

Dear Dr. Gow:

I'm pleased to inform you that your manuscript has been deemed suitable for publication in PLOS ONE. Congratulations! Your manuscript is now with our production department. 

Kind regards, 

on behalf of

Professor Mohammad Asghari Jafarabadi 

Academic Editor

PLOS ONE